

# Berberine-induced browning and energy metabolism: mechanisms and implications

Aslıhan Alpaslan Ağaçdiken and Zeynep Göktaş

Department of Nutrition and Dietetics, Hacettepe University, Ankara, Turkey

## ABSTRACT

Obesity has become a global pandemic. The approaches researched to prevent it include decreasing energy intake and/or enhancing energy expenditure. Therefore, research on brown adipose tissue is of great importance. Brown adipose tissue is characterized by its high mitochondrial content. Mitochondrial uncoupling protein 1 (UCP1) releases energy as heat instead of chemical energy. Thermogenesis increases energy expenditure. Berberine, a phytochemical widely used in Asian countries, has positive effects on body weight control. While the precise mechanisms behind this effect remain unclear, the adenosine monophosphate-activated protein kinase (AMPK) pathway is known to play a crucial role. Berberine activates AMPK through phosphorylation, significantly impacting brown adipose tissue by enhancing lipolytic activity and increasing the expression of UCP1, peroxisome proliferator-activated receptor γ-co-activator-1α (PGC1α), and PR domain containing 16 (PRDM16). While investigating the mechanism of action of berberine, both the AMPK pathway is being examined in more detail and alternative pathways are being explored. One such pathway is growth differentiation factor 15 (GDF15), known for its appetite-suppressing effect. Berberine's low stability and bioavailability, which are the main obstacles to its clinical use, have been improved through the development of nanotechnological methods. This review examines the potential mechanisms of berberine on browning and summarizes the methods developed to enhance its effect.

## INTRODUCTION

Obesity is a growing health concern worldwide, primarily driven by an imbalance between energy intake and expenditure (*Mayoral et al., 2020*). This condition is associated with numerous metabolic disorders, including diabetes, cardiovascular diseases, and non-alcoholic fatty liver disease. Addressing obesity requires innovative approaches to increase energy expenditure while reducing energy storage. Brown adipose tissue (BAT) and the browning of white adipose tissue have emerged as promising therapeutic targets due to their roles in thermogenesis. This process converts stored energy into heat, thereby increasing total energy expenditure (*Marlatt & Ravussin, 2017*).

Berberine is a natural compound extensively utilized in traditional medicine across many Asian countries, especially China (*Wang et al., 2017*). It is an isoquinoline alkaloid derived from several plants, such as *Coptis chinensis*, *Berberis aquifolium*, *Berberis vulgaris*, and *Berberis aristate*. Plants containing berberine have been used since ancient times.

Corresponding author
Zeynep Göktaş, zeynep.
goktas@hacettepe.edu.tr

*Coptis chinensis* was used approximately 2,200 years ago for many health issues, particularly digestive system diseases (*Song, Hao & Fan, 2020*). Approximately 1,500 years ago, Hongjing Tao mentioned the antidiabetic properties of berberine plants in the book "Note of Elite Physicians" (*Zhang et al., 2014a*). With technological advancements, the active component in these plants was identified as berberine, and consequently, the number of studies on berberine has recently increased. Berberine is believed to have numerous effects, including anti-obesity, hypoglycemic, hypolipidemic, hypotensive, and anti-inflammatory effects (*Hesari et al., 2018*; *Pirillo & Catapano, 2015*; *Yarla et al., 2016*).

Berberine is regarded as a potential anti-obesity agent because of its beneficial health effects. Berberine may increase thermogenesis, positively affect carbohydrate and lipid metabolism, suppress appetite, regulate intestinal permeability and hepatic gluconeogenesis, and modulate the microbiota (*Ilyas et al., 2020*; *Park, Jung & Shim, 2020*; *Rong et al., 2021*; *Wu et al., 2019*; *Zhang et al., 2020*, *2014c*). Taking 500 mg of berberine three times a day for 12 weeks may result in an average weight loss of approximately 2.3 kg (5 pounds) in individuals with obesity (*Hu et al., 2012*). Berberine's effect on thermogenesis is one of the most researched topics in this area. Although the exact mechanism is not yet fully understood, berberine induces adipose tissue browning and thermogenesis through various pathways (*Zhang et al., 2015*, *2008*). This browning effect is considered a key mechanism contributing to berberine's potential role in weight loss, as it promotes increased energy expenditure and thermogenesis, addressing the energy imbalance central to obesity. Beyond its metabolic effects, berberine is relatively safe. While berberine toxicity is rarely observed in animals, human studies have reported some mild side effects, such as gastrointestinal disturbances like diarrhea or constipation (*Imenshahidi & Hosseinzadeh, 2019*; *Zhang et al., 2010*). Plants rich in berberine have been reported to be safe, showing no adverse effects on creatinine levels or liver function (*Linn et al., 2012*). The side effects of berberine vary depending on the route of administration, dosage, and duration of use.

This review aims to clarify the mechanisms of adipose tissue browning, which are essential for preventing obesity and its associated conditions, as well as to examine the effects of berberine on these mechanisms. There are different reviews in the literature examining the health effects of berberine. However, the number of articles presenting up-to-date data on the effects of berberine on adipose tissue browning and BAT activation is limited. Due to its low bioavailability, most studies on berberine are *in vitro*. Methods that could address this issue are provided in this review.

## SURVEY METHODOLOGY

In this review, articles containing the keywords "berberine" along with "brown adipose tissue," "browning," and "thermogenesis" in their titles or abstracts were searched in the PubMed, Science Direct, and Scopus databases. Since most of the relevant sources were recently published, no year restriction was applied. Only articles in English are considered. Research articles and reviews were included. A search using these criteria resulted in 278 research articles. After applying search filters, the titles of the resulting articles were reviewed first, followed by their abstracts. Articles with abstracts relevant to the research

topic were then examined in detail. Studies that met the search criteria but were not relevant to the topic, did not provide sufficient data, or were inaccessible in full text were excluded. As a result, 10 studies relevant to the purpose of this review were included. Limiting the search to articles in English resulted in the exclusion of studies written in native languages from Asian countries, where the use of berberine is more prevalent. This can be considered a limitation of this study.

## The audience it is intended for

This review may attract the attention of experts particularly interested in phytochemicals and adipose tissue, those investigating methods used to combat obesity, and those with an interest in nanotechnological approaches. With a better understanding of berberine's effects on adipose tissue browning and the underlying mechanisms, it can be considered a potential drug for obesity prevention and/or treatment.

# BERBERINE AND ITS PHARMACOKINETIC PROPERTIES

Berberine (2,3-methylenedioxy-9,10-dimethoxyprotoberberine chloride) is yellow, odorless, and has a bitter taste (*Feng et al., 2019*). It is more soluble in organic solvents and has low water solubility. Its molecular weight is 336.36 g/mol. It can be extracted from its source plants, or it can be synthesized (*Feng et al., 2019*).

## Absorption and bioavailability

While the health effects of berberine are intriguing, its low oral bioavailability (approximately 5%) is well-documented (*Habtemariam, 2020*; *Wang et al., 2017*). One reason for its limited bioavailability is its high binding affinity for plasma proteins (*Mirhadi, Rezaee & Malaekeh-Nikouei, 2018*). Therefore, research has focused on the metabolites of berberine and their health effects. Clinical evaluations have shown that intravenous administration of berberine increases its concentration in the blood (*Han et al., 2021*). However, this increase can dangerously lower blood pressure, potentially leading to death. Therefore, oral intake is safer than intravenous administration in clinical applications.

## Distribution

The distribution of berberine varies depending on its form and route of administration. With oral administration, tissue distribution is high, while plasma concentration is relatively low (*Tan et al., 2013*). It particularly accumulates in the liver, adipose tissue, kidneys, and muscles. In intravenous administration, tissue distribution is faster (*Liu et al., 2010*). While this is desirable in acute conditions, it is not practical for chronic use. Intraperitoneal administration offers higher bioavailability compared to the oral route, but tissue distribution is slower compared to intravenous administration. Four hours after administration, berberine levels in many tissues are about 70 times higher than in plasma (*Han et al., 2021*). However, berberine levels remain stable in certain tissues such as the liver and muscles. Encapsulation of berberine or co-administration with P-glycoprotein inhibitors enhances its absorption and improves tissue distribution (*Imenshahidi & Hosseinzadeh, 2019*; *Liu et al., 2010*).

## Metabolism

Berberine administered orally undergoes primary metabolism in the liver and intestines. The liver enzymes responsible for metabolizing berberine include cytochrome (CY) P2D6 and CYP1A2 subtypes of CYP450 (Fig. 1) (*Li et al., 2011*). *In vivo* studies showed that the primary metabolic pathways of berberine include demethylation, demethylenation, reduction, and hydroxylation (*Liu et al., 2009*). These processes lead to phase 1 metabolites of berberine. Phase 2 metabolites form through the conjugation of these metabolites with sulfuric acid or glucuronic acid.

Berberine also undergoes metabolism in the intestines, where its structure and content can be altered by intestinal flora (*Han et al., 2021*). This alteration occurs through demethoxylation and hydrogenation pathways, involving nitroreductases produced by intestinal flora. Dihydroberberine, a form that can be absorbed in the intestines, is produced through hydrogenation. After absorption, this form oxidizes back to berberine and enters circulation (*Han et al., 2021*).

Berberine is metabolized into four primary metabolites: berberrubine, thalifendine, demethyleneberberine, and jatrorrhizine (*Hu et al., 2018*). After oral ingestion, berberine is distributed throughout the body, including the small intestine (undergoing presystemic elimination), liver (where it accumulates), kidneys, muscles, heart, and pancreas. The primary metabolic pathways include oxidative demethylation leading to the production of berberrubine, followed by glucuronidation. After intravenous administration, berberine undergoes oxidative demethylation, resulting in the production of demethyleneberberine, followed by glucuronidation of demethyleneberberine (*Hu et al., 2018*).

## Excretion

Berberine is primarily excreted *via* urine, feces, and bile (*Ma et al., 2013*). Due to enterohepatic circulation, excretion *via* bile is slow. The excretion of berberine varies depending on the route of administration (*Han et al., 2021*). In rats, oral or gavage administration of berberine results in feces being the primary route of excretion, with the excreted form remaining unchanged as berberine (*Feng et al., 2020*). Excretion *via* urine and bile is minimal and primarily in the form of berberine metabolites. Intravenous administration of berberine shows urine as the primary excretion route (*Feng et al., 2020*).

## TYPES OF ADIPOSE TISSUE AND BROWNING

Adipose tissue consists of white adipose tissue (WAT) and BAT, composed mainly of white and brown adipocytes, respectively (*Kurylowicz & Puzianowska-Kuznicka, 2020*). The origins, morphologies, anatomical locations, and nearly all functions of these two types are different from each other (Table 1). White adipocytes consist of a single large lipid droplet with a non-centrally located nucleus, and very few mitochondria (*Bargut et al., 2017*). Brown adipocytes contain many small lipid droplets, have a centrally located nucleus and are dark in color due to the high number of mitochondria. Adipocyte precursor cells, also known as adipose stem cells, can differentiate into white, beige, or brown adipocytes (*Xue et al., 2015b*). The expression of myogenic factor-5 determines the difference between white and brown adipocytes. Myogenic factor-5 is associated with

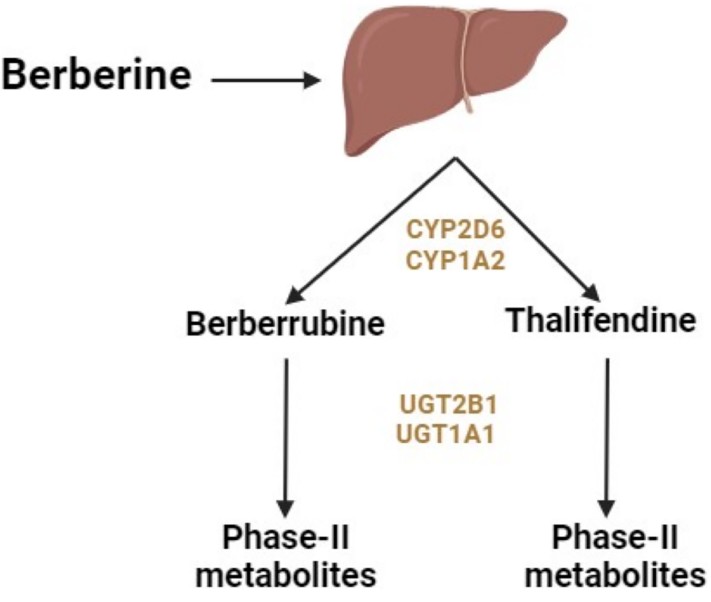

**Figure 1** **Metabolism of berberine in the liver.**

thermogenic activities and present in brown adipocyte precursors but not in white adipocytes (*Xue et al., 2015b*).

White adipose tissue begins to develop during the second trimester of pregnancy, and BAT starts to develop towards the end of the second trimester (*Cypess, 2022*). In newborns, both WAT and BAT are fully developed to perform their functions. WAT is categorized into two main types: visceral and subcutaneous and its primary function is to store energy. Energy stored in the form of triglycerides undergoes lipolysis when required, resulting in the release of fatty acids as fuel. Fifty years ago, it was thought that BAT, which was known to be present in infants, was not present in adults due to insufficient imaging techniques. With the advancement of positron emission tomographic and computed tomographic (PET/CT) imaging technique, active BAT was initially observed in the supraclavicular region of adult humans (*Cypess et al., 2009*). Later, the presence of BAT was also identified in the neck, axillary, abdominal, and paraspinal regions in adults (*Keuper & Jastroch, 2021*). There is a high concentration of mitochondria in brown adipocytes. Uncoupling protein 1 (UCP1) in the inner membrane of mitochondria is essential for browning and thermogenesis mechanisms (*Kurylowicz & Puzianowska-Kuznicka, 2020*). Uncoupling protein 1 releases energy as heat instead of chemical energy by uncoupling mitochondrial respiration from adenosine triphosphate (ATP) synthesis. Therefore, it facilitates thermogenesis and promotes energy expenditure. Consequently, the identification of BAT in adults represents a promising avenue for combating the obesity epidemic. Additionally, "beige/brite" adipocytes are morphologically resemble white adipocytes but exhibit brown adipocyte function under adequate stimuli (*Cheng et al., 2021*). Phytochemicals such as berberine, resveratrol, and curcumin, and dietary components like fish oil and retinoic acid, along with cold exposure, exercise, and β-adrenergic factors, stimulate beige adipocytes through a process known as "browning" (*Cheng et al., 2021*; *Okla et al., 2017*).

**Table 1 Characteristics of different adipocytes.**

|  | White | Beige | Brown |
|---|---|---|---|
| Anatomical location | Subcutaneous, visceral | White depots and supraclavicular | Interscapular, adrenal, neck |
| Morphology | Unilocular | Multilocular | Multilocular |
| Lipid droplets | Large | Numerous and small | Numerous and small |
| Progenitor | Pdgfr-α | Pdgfr-α | Myf5+ |
| Main function | Energy storage | Thermogenesis | Thermogenesis |
| Mitochondrial biogenesis | Low | Medium | High |

**Note:**
Pdgfr-α, platelet-derived growth factor receptor alpha; Myf5+, myogenic factor 5-positive.

As browning progresses, beige adipocytes, which have a morphology similar to white adipocytes, begin to perform functions similar to brown adipocytes. As mitochondrial biogenesis and UCP1 expression increase, energy expenditure through thermogenesis will also increase. This is very important in combating obesity, which is mainly caused by an imbalance between energy intake and expenditure. The beige adipocytes lose their brown characteristics and return to the characteristics of the white adipocytes when the stimulus is removed (*Ziqubu et al., 2023*). This process, called "whitening," is considered the opposite of browning. It can be seen in obesity and during aging (*Graja, Gohlke & Schulz, 2019*; *Ziqubu et al., 2023*).

## Energy expenditure and thermogenesis

Obesity typically arises when energy intake exceeds energy expenditure (*Lin & Li, 2021*). To prevent or treat obesity, energy intake must be reduced and/or energy expenditure must be increased. Dietary interventions are employed to reduce energy intake. To increase energy expenditure, it is crucial to understand the components of total energy expenditure. Approximately 70% comes from the resting metabolic rate, including the thermic effect of the foods, which reflects the energy used for digestion, absorption, and processing of nutrients (*Tran et al., 2022*). Twenty percent is attributed to energy expenditure from physical activity, divided into non-exercise activity thermogenesis and exercise-induced thermogenesis. Ten percent arises from diet-induced thermogenesis, which occurs in response to excess caloric intake. Lastly, cold-induced thermogenesis is variable and involves mechanisms like shivering thermogenesis and non-shivering thermogenesis (*Saito et al., 2020*). Among these components, diet-induced thermogenesis and non-shivering thermogenesis are primarily mediated by BAT (*Tran et al., 2022*). This is because thermogenesis is primarily associated with mitochondria and UCP1, and BAT has a high mitochondrial content (*Van Thi-Tuong, Van Vu & Van Pham, 2023*). An increase in BAT is expected to enhance these components and, consequently, increase overall energy expenditure.

## BROWNING MECHANISMS AND BAT ACTIVATION

There are two ways to increase thermogenesis through adipose tissue. The first is to increase browning and the second is to increase the already existing BAT activation.

Browning can occur in two ways (*Kurylowicz & Puzianowska-Kuznicka, 2020*). The first is by differentiating from precursor/stem cells and the second is by transdifferentiating from mature adipocytes. Subcutaneous adipocytes are more likely to brown than visceral adipocytes due to their ability to differentiate (*Gustafson & Smith, 2015*).

Beta-adrenergic receptor activation is considered a key stimulus for browning. The receptors involved in this system may vary between species. For example, in rodents, the β-3 adrenergic receptor (β3-AR) is involved in browning, whereas in humans the β2-AR is involved (*Blondin et al., 2020*). Cold exposure, similar to β-adrenergic agonists, activates the sympathetic nervous system and releases norepinephrine. When the β-adrenergic receptor is stimulated, it activates cyclic adenosine monophosphate (cAMP) and protein kinase A (PKA).

Protein kinase A activates cAMP response element-binding protein (CREB), p38 mitogen-activated protein kinase (p38-MAPK), and mechanistic target of rapamycin (mTOR) phosphorylation. cAMP response element-binding protein and p38-MAPK increase the transcription of peroxisome proliferator-activated receptor γ-co-activator-1α (PGC-1α), which activates transcription factors that induce mitochondrial biogenesis (*Deng et al., 2019*). Mechanistic target of rapamycin is also important for mitochondrial biogenesis (*Wei et al., 2015*).

Sympathetic activation is central to the complex mechanisms that lead to mitochondrial biogenesis, browning, and thermogenesis. Current research highlights several stimuli that enhance these processes, including exercise, specific dietary components, and pharmacological agents. For instance, exercise stimulates the production and release of irisin, which activates the p38-MAPK and extracellular signal-regulated kinase (ERK) pathways, thereby increasing UCP1 expression (*Zhang et al., 2014b*). Exercise also increases the expression of fibroblast growth factor-21 (FGF21) in the liver and adipose tissue. Its increase in adipose tissue induces UCP1 expression in white adipocytes (*Reilly et al., 2021*).

Additionally, exercise-induced reactive oxygen species (ROS) and its effects on the nervous system also play a role in adipose tissue browning (*Mu et al., 2021*). Chronic administration of β-adrenergic agonists and leptin increases sympathetic innervation and stimulates thermogenesis (*Jimenez et al., 2003*; *Wang et al., 2020*).

Other factors also play a role in promoting browning. For example, the lipid-lowering agent fenofibrate increases thermogenesis by activating peroxisome proliferator-activated receptor (PPAR)-α (*Rachid et al., 2015*). Similarly, PPAR agonists, agents that activate the Adenosine monophosphate (AMP)-activated protein kinase (AMPK) pathway, and substances such as nicotine (but not smoking) can stimulate browning by promoting mitochondrial biogenesis (*Gaidhu et al., 2009*; *Yoshida et al., 1999*).

Examining the expression and/or protein levels of transcription factors is one of the primary methods used to assess browning. These factors interact with each other, influencing adipogenesis and browning (Table 2). Among these markers, UCP1 is considered a definitive indicator of browning and thermogenic activity. Uncoupling protein 1 activity is regulated by free fatty acids, which enhance its activity, and purine nucleotides, which inhibit it (*Macher et al., 2018*). Which regulatory protein binds to the

**Table 2 Some transcription factors in browning.**

| Transcription factor | Role in browning/thermogenesis | Interactions |
|---|---|---|
| UCP1 | Key marker of browning and thermogenesis. Activates heat production in adipocytes. | Regulated by free fatty acids (activates) and purine nucleotides (inhibits). Transcription regulated by various factors (*e.g.*, PRDM16, PPARγ) (*Macher et al., 2018*; *Jash et al., 2019*). |
| PPARγ | Regulates both fat and carbohydrate metabolism. Plays a role in adipogenesis and lipid storage. Can induce browning under certain conditions. | Interacts with LXR and RIP140 to downregulate UCP1. Agonists increase insulin sensitivity and browning but can also increase adiposity (*Machado et al., 2022*; *Wang et al., 2008*). |
| PGC-1α | Key factor for mitochondrial biogenesis. Stimulates thermogenesis in muscle and brown adipocytes. | Activated by β-adrenergic receptor stimulation. Regulates UCP1 and other thermogenic genes. Stimulated by exercise, cold, and pharmacological agents (*Deng et al., 2019*; *Ishibashi & Seale, 2015*). |
| CIDEA | Prevents downregulation of UCP1, thus promoting browning and thermogenesis. | Inhibits LXR to prevent UCP1 downregulation (*Jash et al., 2019*). |
| PRDM16 | Activates thermogenic genes in WAT. Essential for the browning of subcutaneous WAT. | Stimulates PGC-1α expression and is necessary for maintaining beige adipocytes. Low PRDM16 expression can reverse browning (*Harms et al., 2014*; *Ishibashi & Seale, 2015*). |

**Note:**
CIDEA, Cell Death-Inducing DNA Fragmentation Factor-Like Effector A; LXR, Liver X receptor; PGC-1α, Peroxisome Proliferator-Activated Receptor γ Co-Activator 1α; PPARγ, Peroxisome Proliferator-Activated Receptor γ; PRDM16, PR Domain Containing 16; RIP140, receptor-interacting protein 140; UCP1, Uncoupling Protein 1; WAT, white adipose tissue.

gene determines the transcriptional regulation of UCP1 (*Villarroya, Peyrou & Giralt, 2017*). In the absence of UCP1, lipogenesis and liver steatosis increase (*Winn et al., 2017*).

PPARγ-co-activator-1α is one of the most effective factors in stimulating mitochondrial biogenesis in both muscle and brown adipocytes (*Deng et al., 2019*). Another important browning factor is cell death-inducing DNA fragmentation factor-like effector A (CIDEA), which prevents the downregulation of UCP1 by inhibiting liver X receptors (LXRs) (*Jash et al., 2019*). PR domain containing 16 (PRDM16) can activate thermogenic genes in WAT (*Ishibashi & Seale, 2015*). It activates PGC-1α and is necessary for the browning of subcutaneous WAT. Low expression of PRDM16 can reverse browning and convert beige adipocytes back to white adipocytes (*Harms et al., 2014*). PR domain containing 16 is thus critical for maintaining beige adipocytes and their thermogenic activity.

Peroxisome proliferator-activated receptor-γ is another key transcription factor, influencing both fat and carbohydrate metabolism. It interacts with LXR and receptor-interacting protein 140 (RIP140) to downregulate UCP1 (*Wang et al., 2008*). Applying PPARγ agonists can increase insulin sensitivity and browning but may also increase visceral adiposity and unwanted body weight (*Machado et al., 2022*). Therefore, PPARγ plays a crucial role in both browning and whitening processes.

Cold exposure increases browning partly by enhancing noradrenergic stimulation, which increases iodothyronine deiodinase-2 (DIO2), converting thyroxin (T4) to triiodothyronine (T3) (*Kurylowicz & Puzianowska-Kuznicka, 2020*). Elevated T3 levels stimulate the sympathetic nervous system, thereby increasing UCP1 expression. Fibroblast growth factor-21 increases UCP1 expression by upregulating PGC-1α (*Fisher et al., 2012*). It also enhances browning by increasing intracellular $Ca^{++}$ levels. Forkhead box C2

(FoxC2), which is expressed in adipose tissue, mediates a thermogenic effect *via* the PKA pathway by increasing the expression of PGC-1α and UCP1 (*Kajimura, Seale & Spiegelman, 2010*).

Both browning and the activation of brown adipose tissue are triggered by similar stimuli (*Kurylowicz & Puzianowska-Kuznicka, 2020*). The method used to determine BAT activation is 2-deoxy-2-[$^{18}$F] fluoro-D-glucose ([$^{18}$F]FDG)-PET/CT imaging. This imaging technique allows for the tracking of the presence and size of brown adipose tissue (*van der Lans et al., 2014*).

## EFFECTS OF BERBERINE ON BROWNING AND BAT ACTIVATION

A significant portion of berberine's effects on browning and BAT activation occurs *via* the AMPK pathway. Beyond this long-studied area, new pathways are being explored, with recent attention on growth differentiation factor 15 (GDF15). The effects of berberine on brown adipose tissue are shown in Fig. 2. Studies investigating the effects of berberine on browning are summarized in Table 3.

### Adenosine monophosphate-activated protein kinase

AMP-activated protein kinase, a serine/threonine kinase, is one of the key regulators of energy metabolism (*Wu & Zou, 2020*). Especially in critical situations such as insufficient energy intake, it binds to adenosine diphosphate (ADP) or AMP and regulates key enzymes and proteins in carbohydrate, protein, and lipid metabolism. One of its important targets is PGC1α, which plays a role in mitochondrial homeostasis by converting type IIb muscle fibers into type I and type II fibers that contain more mitochondria (*Jager et al., 2007*). AMP-activated protein kinase is one of the important factors regulating mitochondrial biogenesis in adipocytes and other tissues (*Yang et al., 2016*). Mitochondrial biogenesis ensures the production of ATP that meets the increased energy expenditure. Through these mechanisms, AMPK ensures energy homeostasis by increasing ATP production and/or reducing its consumption (*van der Vaart, Boon & Houtkooper, 2021*).

The activation of AMPK is linked to browning in adipose tissue. When factors stimulating the beta-adrenergic receptor decrease, BAT activation is reduced, leading to decreased AMPK phosphorylation in BAT. AMP-activated protein kinase is influenced by triggers that activate the beta-adrenergic system, such as cold exposure (*Mulligan et al., 2007*). Lipases play an important role in linking AMPK to brown adipose tissue activation/browning (*van der Vaart, Boon & Houtkooper, 2021*). AMP-activated protein kinase increases the phosphorylation of hormone-sensitive lipase (HSL) and adipose triglyceride lipase (ATGL), regulates lipoprotein lipase (LPL) and cluster of differentiation 36 (CD36), and stimulates acetyl-CoA carboxylase (ACC) to remove the suppression of carnitine palmitoyltransferase 1 (CPT1). All these actions lead to an increase in fatty acids or their entry into the mitochondria. The fatty acids bind to UCP1, inducing thermogenesis. It also increases UCP1 expression by enhancing PPARγ deacetylation through the AMPK/Sirtuin 1 (SIRT1) pathway (*Xu et al., 2021*). In addition to activating UCP1, AMPK also stimulates

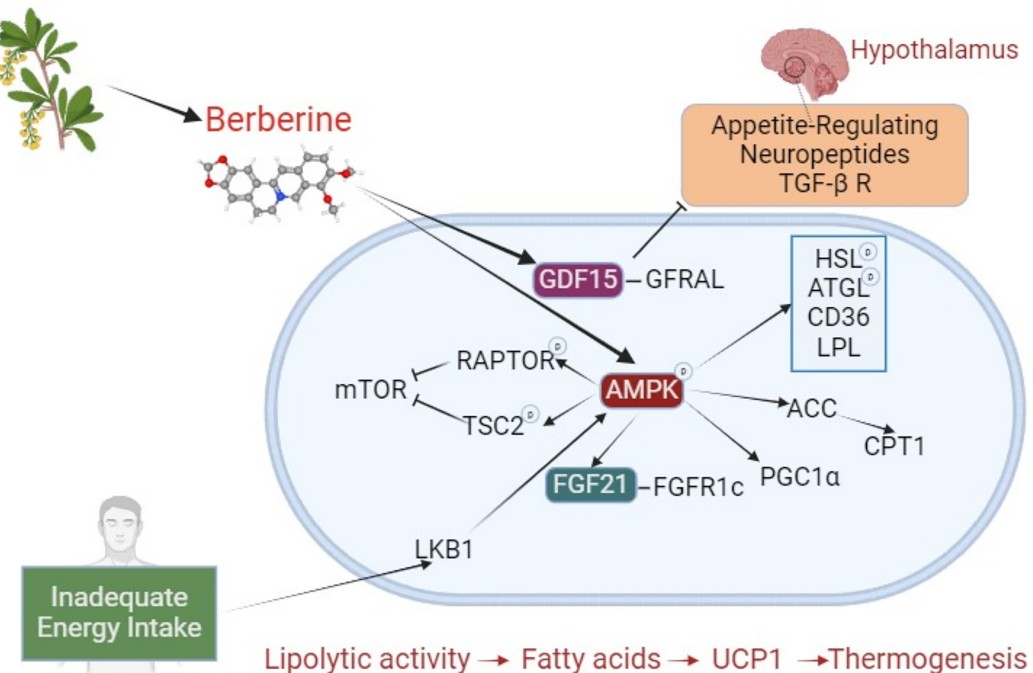

**Figure 2 The effect of berberine on adipose tissue browning.** Berberine derived from plant sources phosphorylates AMPK, inducing HSL, ATGL, LPL, and CD36, and stimulates lipolysis. Through ACC activation, it prevents CPT1 inhibition. By increasing FGF21 expression, it enhances binding to the FGFR1c receptor. These interactions, which result in increased lipolysis and fatty acids, initiate a mechanism that leads to thermogenesis with UCP1. During inadequate energy intake, LKB1 activates AMPK, which phosphorylates RAPTOR and TSC2, suppressing mTOR (negative impact on BAT activation). By stimulating PGC1α, it plays a crucial role in browning. Independently of AMPK, berberine increases GDF15, promoting its binding to the GFRAL receptor. GFRAL stimulates appetite-related neuropeptides and the TGF- β R in the hypothalamus. The TGF- β R is a regulator of precursor cells that promote browning. Abbreviations: ACC, acetyl-CoA carboxylase; AMPK, AMP-activated protein kinase; ATGL, adipose triglyceride lipase; BAT, brown adipose tissue; CD36, cluster of differentiation 36; CPT1, carnitine palmitoyltransferase 1; FGF21, fibroblast growth factor 21; FGFR1c, fibroblast growth factor receptor 1c; GDF15, Growth differentiation factor 15; GFRAL, glial cell-derived neurotrophic factor family receptor alpha-like; HSL, hormone sensitive lipase; LKB1, liver kinase B1; mTOR, mechanistic target of rapamycin; LPL, lipoprotein lipase; PGC1α, PPARγ-co-activator-1α; RAPTOR, regulator-y-associated protein of mTOR; TGF- β R, transforming growth factor β receptors; TSC2, tuberous sclerosis complex 2; UCP1, uncoupling protein 1. Created with BioRender.com.

PRDM16 and PPARγ by reducing isocitrate dehydrogenase 2 (IDH2) and α-ketoglutarate (*Yang et al., 2016*).

Furthermore, berberine increases FGF21 expression *via* the AMPK pathway, enhancing binding to the FGFR1c receptor. This stimulates both lipolysis and the activation of PGC1α and UCP1 (*Fisher et al., 2012*; *Hirai et al., 2019*). When energy intake is limited, liver kinase B1 (LKB1) is activated and upregulates AMPK (*Agarwal et al., 2015*). AMP-activated protein kinase phosphorylates regulatory-associated protein of mTOR (RAPTOR) and tuberous sclerosis complex 2 (TSC2), thereby suppressing mTOR. For brown adipocyte differentiation, mTOR is crucial. Therefore, it can be inferred that the activation of AMPK does not have a positive effect on BAT development. However, AMPK

**Table 3 Studies investigating the effects of berberine on browning.**

| | Dose | Duration | Suggested Pathway | Results | References |
|---|---|---|---|---|---|
| 3T3-L1 preadipocytes | 0.5, 1, 5, 10 μM | 7 days | cAMP/PKA | Adipogenic genes (C/EBP-α, PPARγ) ↓, CREB activity↓ | Zhang et al. (2015) |
| HepG2 cells | 5, 10, 15 μM | N/A | AMPK | AMPK phosphorylation↑, ACC↑, fatty acid oxidation↑ | Brusq et al. (2006) |
| Male Syrian golden hamsters | 100 mg/kg/day | 10 days | | | |
| Male C57BL/6J mice | 50 and 100 mg/kg/day | 14 days | GDF15 | Serum GDF15↑, GFRAL↑, appetite↓, UCP1↑ | Li et al. (2023) |
| Obese C57BLKS/J-Lepr[db]/Lepr[db] male mice and wild-type mice | 5 mg/kg/day | 4 weeks | AMPK- PGC1α | Energy expenditure↑, weight gain↓, BAT activity↑, UCP1↑, PGC-1α↑, CIDEA↑ | Zhang et al. (2014c) |
| Male C57BL/6J mice | 100 mg/kg/day | 10 weeks | AMPK | AMPK↑, complex I↓, AMP/ATP↑, ADP/ATP↑ | Turner et al. (2008) |
| Male Wistar rats | | 4 weeks | | | |
| Female Sprague-Dawley rats | 380 mg/kg/day | 2 weeks | N/A | Olanzapine-induce BAT loss↓, weight gain↓, adiposity↓, AMPK↑, UCP1↑, PGC1α↑. Food intake did not change. | Hu et al. (2014) |
| Male C57BL/6J mice | 25 and 100 mg/kg/day | 12 weeks | AMPK-SIRT1 | Distribution of BAT↑, thermogenesis↑, body weight↓, AMPK/SIRT1 activation↑, PPAR↑ deacetylation↑, UCP1 expression↑ | Xu et al. (2021) |
| Obese C57BLKS/J-Lepr[db]/Lepr[db] male mice | 5 mg/kg/day | 26 days | AMPK | Lipogenesis (FAS, PPARγ)↓, expression of browning markers (PGC1α)↑, AMPK activation↑, body weight↓. Food intake did not change. | Lee et al. (2006) |
| Wistar rats | 380 mg/kg/day | 2 weeks | | | |
| Female Sprague-Dawley rats | 380 mg/kg/day | 2 weeks | N/A | Blood lipid levels↓, weight loss↑ | Hu et al. (2012) |
| Obese humans | 1.5 g/day | 12 weeks | | | |
| Male C57BL/6J mice | 1.5 mg/kg/day | 6 weeks | AMPK–PRDM16 | Both in mice and humans: Brown adipocyte differentiation↑, PRDM16 transcription↑ | Wu et al. (2019) |
| NAFLD patients | 1.5 g/day | 1 month | | In mice: thermogenesis↑, energy expenditure↑ AMPK is essential for the browning effect of berberine. | |

**Note:**

ACC, acetyl-CoA carboxylase; ADP, adenosine diphosphate; AMP, adenosine monophosphate; AMPK, AMP-activated protein kinase; ATP, adenosine triphosphate; BAT, brown adipose tissue; cAMP, cyclic adenosine monophosphate; C/EBP-α, CCAAT/enhancer-binding protein alpha; CIDEA, cell death-inducing DNA fragmentation factor-like effector A; CREB, cAMP response element-binding protein; FAS, fatty acid synthetase; GDF15, Growth differentiation factor 15; GFRAL, glial cell-derived neurotrophic factor family receptor alpha-like; PGC1α, PPARγ-co-activator-1α; PKA, protein kinase A; PPARγ, Peroxisome proliferator-activated receptor-γ; PRDM16, PR domain containing 16; SIRT1, Sirtuin 1; UCP1, uncoupling protein 1.

is important in regulating browning (*Perdikari et al., 2018*). This is indicated by the inability to brown in the absence of AMPK. AMP-activated protein deficiency prevents brown adipocyte maturation, with the key subunit involved being AMPK-α1 (*Perdikari et al., 2017*). However, α2 or combined knockdown of other subunits (β1, β2, γ1, and γ3) also prevents browning by reducing UCP1. While studies are showing that berberine is an activator of AMPK, it has been shown specifically to bind to the γ-subunit and activate the

AMPK-α-ketoglutarate-PRDM16 pathway (*Garcia & Shaw, 2017*; *Hardie, 2013*; *Wu et al., 2019*). Other mechanisms include the activation of AMPK by berberine through inhibition of complex I, thereby increasing the AMP:ATP and ADP:ATP ratios (*Turner et al., 2008*).

### Growth differentiation factor 15

Growth differentiation factor 15 (GDF15) is a cellular stress biomarker (*Li et al., 2023*). Growth differentiation factor 15 expression is negatively associated with appetite and food intake in diet-induced obese mice. Some drugs, like metformin, promote weight loss by increasing GDF15 levels as one of their mechanisms of action (*Coll et al., 2020*). GDF15 binding to its receptor, glial cell-derived neurotrophic factor family receptor alpha-like (GFRAL) leads to reduced appetite in the hypothalamus by inhibiting specific neuropeptides and transforming growth factor-β receptors (TGF-β-R) (*Wang et al., 2021*; *Yang et al., 2017*). Transforming growth factor β receptors are regulators of precursor cells that promote browning (*Wankhade et al., 2018*). In a recent *in vitro* study, berberine was shown to increase GDF15 expression in adipocytes (*Li et al., 2023*). Subsequently, when berberine was administered by gavage to obese mice, the circulating levels of GDF15 increased, appetite and food intake decreased, and as a result, the mice lost weight. Although there are studies suggesting that GDF15 can stimulate browning, thermogenesis, and energy expenditure by increasing UCP1 expression, it has also been observed that GDF15 does not affect or even downregulate UCP1 (*Choi et al., 2020*; *Chrysovergis et al., 2014*; *Li et al., 2023*). While GDF15 secretion from adipocytes is normally quite low, berberine intake significantly increased this secretion, particularly in BAT (*Li et al., 2023*; *Wang et al., 2021*). Although the effect of increasing GDF15 on UCP1 is not definitive, increasing BAT mass may enhance berberine-induced GDF15 secretion, thereby helping to control body weight.

## FUTURE IMPLICATIONS

Berberine chloride and sulfate salts are more soluble, but the clinical use of the free form is limited due to its hydrophobic nature, low gastrointestinal absorption, and rapid metabolism. Different strategies have been developed to address the issue of low bioavailability in phytochemicals like berberine (*Li et al., 2021*; *Mirhadi, Rezaee & Malaekeh-Nikouei, 2018*; *Qiao et al., 2018*; *Yu et al., 2017*). These methods can be broadly categorized into three main approaches. These include:

- Changing the administration method (nanotechnology methods)
- Altering the chemical structure (organic acid salts of berberine)
- Co-administration with P-glycoprotein inhibitors (*e.g.*, glycine, cyclosporine A)

Among these methods, nanotechnology methods are the most used, especially in clinical studies. Nanoparticles are particles with diameters ranging from 10 to 1,000 nm. With these methods, particle size is reduced as much as possible, surface properties are optimized, and the biologically active material is released at an optimal level. This structure, also known as a nano-carrier, helps berberine reach target tissues while preserving its properties. Nano-carriers can include materials such as micelles,
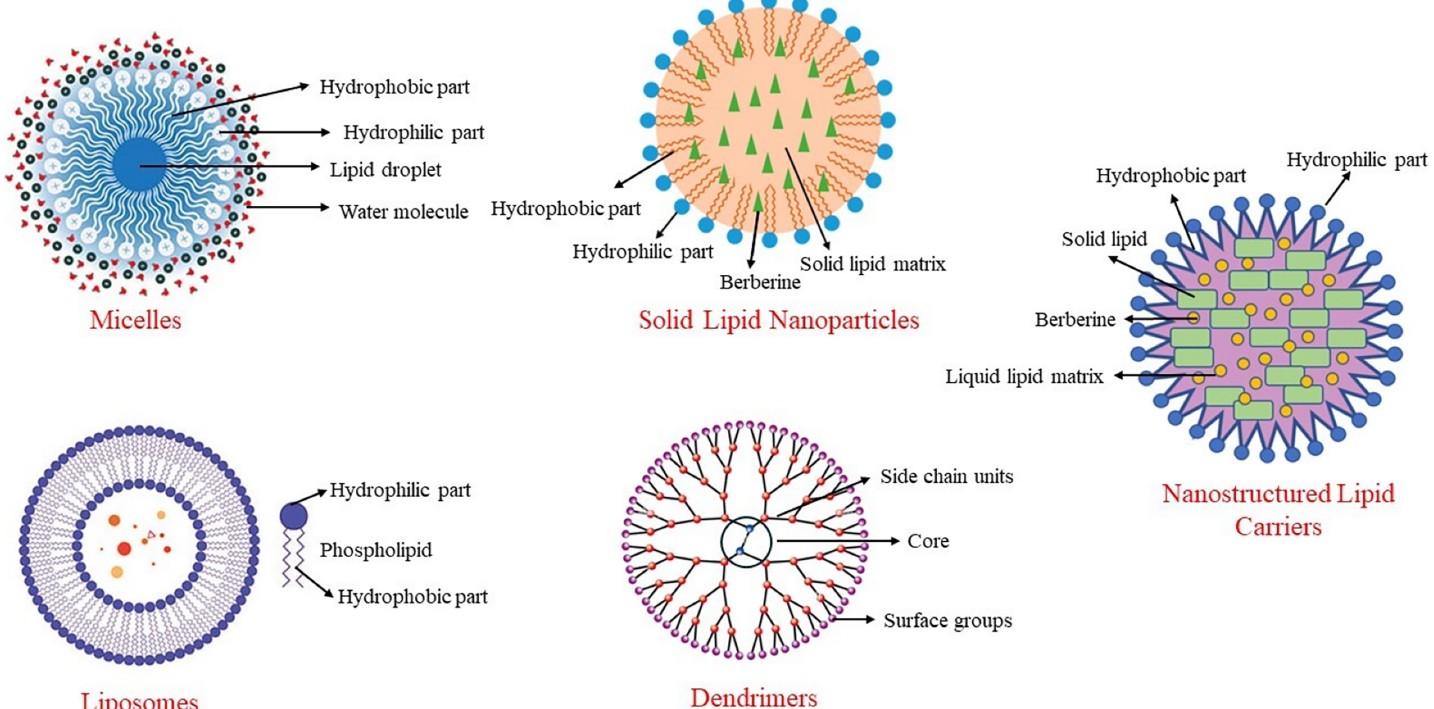

**Figure 3 Some nanoencapsulation methods to improve berberine's bioavailability.**

carbon-based compounds, liposomes, and polymers (*Behl et al., 2022*). Some nanoencapsulation methods used to enhance the bioavailability of berberine are shown in Fig. 3.

## Solid lipid nanoparticles

In a system developed to increase the bioavailability of berberine and extend its duration of action, berberine is transported within solid lipid nanoparticles (*Xue et al., 2015a*). When berberine-SLN was administered orally to db/db mice, berberine was stored in the brain, liver, and jejunum. In this system, the elimination of berberine from the body was reduced, and its circulating levels were increased. Berberine encapsulated in SLNs reduced triglyceride and alanine transaminase (ALT) concentrations in the liver, and its anti-diabetic effect was enhanced through nanoencapsulation. Additionally, the presence of berberine in the brain demonstrated that SLN-encapsulated berberine could cross the blood-brain barrier. Berberine-SLN was spherical. The efficiency of encapsulation was 58%, with a loading capacity of 4.2%. The particle size measured 76.8 nm, and the zeta potential was 7.87 mV. The bioavailability of orally administered berberine-SLN (50 mg/kg body weight) was higher compared to free berberine. In the study conducted by *Xue et al. (2013)*, the peak plasma concentration of free berberine was reported as 11.1 ± 6.24, whereas that of berberine-SLNs was significantly higher at 44.651 ± 4.77. Similarly, the area under the curve (AUC) values were 56.5 ± 29.61 for free berberine and 113.6 ± 72.93 for berberine-SLNs, indicating a substantial improvement in bioavailability with the SLN formulation (*Xue et al., 2013*). Berberine chloride-loaded SLNs are used in studies aimed at

preventing and treating various health issues, including cancer therapy (*Wang et al., 2014*; *Xue et al., 2013*).

## Nanostructured lipid carriers

This nano-carrier method was developed to address the limitations of SLNs by replacing some of the solid lipids in the structure with liquid lipids. This modification increased the loading capacity and prevented berberine leakage during storage. Berberine-NLCs are also spherical, with an encapsulation efficiency of 88%, a particle size of 186 nm, a zeta potential of −36.86 mV, and a polydispersity index of 0.108. Berberine-NLC structures are frequently encountered in studies related to liver health, cognitive functions, and various tumors (*Gendy et al., 2022*; *Raju et al., 2021*).

## Liposomes

Liposomes are nano-carriers composed of cholesterol and phospholipids. Like other lipid-based nanoencapsulation methods, they are spherical. Due to the hydrophobic and hydrophilic properties of phospholipids, liposomes are used to deliver antibacterial, antifungal, anticancer, and anti-inflammatory drugs, as well as phytochemicals (*Akbarzadeh et al., 2013*). The literature contains studies on the use of liposomal berberine for liver diseases, cardiovascular diseases, and certain tumors (*Allijn et al., 2017*; *Calvo et al., 2020*; *Lin et al., 2013*).

## Micelles

Micelles are complex structures based on surfactants that use various phosphatidylcholine mixtures. They can be spherical or resemble a disk. Encapsulating berberine in anhydrous reverse micelles (ARM) increases berberine's oral bioavailability by 2.4 times (*Wang et al., 2011*). Berberine-loaded micelles were found to increase berberine solubility by 800% and its absorption by 364%. Additionally, the efflux rate of berberine within the micelles decreased from 7.54 to 1.05. This indicates that the inhibition of P-glycoprotein-mediated efflux leads to an increase in the intestinal absorption of berberine (*Kwon et al., 2020*).

## Dendrimers

Dendrimers have a branched structure and are polymeric nano-carriers with many functional groups on their surfaces (*Sherje et al., 2018*). Due to these characteristics, dendrimers exhibit high efficacy and bioavailability. Their unique branched structure, high solubility in water, ability to neutralize various toxins, antigens, or microorganisms, and the simplicity of their production method make them particularly useful in the field of pharmacology. Berberine encapsulated in polyamidoamine (PAMAM) dendrimers increases its permeability, enhancing its bioavailability and therapeutic effects (*An et al., 2023*). These dendrimers are biocompatible and safe, making them suitable for use in various medical applications (*Gupta et al., 2017*). They are commonly employed in cancer research, where they help improve drug delivery by targeting cancer cells more effectively and providing controlled release (*Yadav, Semwal & Dewangan, 2023*).

## CONCLUSIONS

Berberine, a herbal compound used in Asia for centuries, has recently gained attention for its health benefits, particularly in weight management. While studies generally report positive effects, the underlying mechanisms remain unclear. Key mechanisms include AMPK pathway activation, increased browning markers like UCP1, and appetite-regulating markers such as GDF15. Given the importance of adipose tissue browning and BAT activation in preventing obesity, berberine's potential to enhance energy expenditure is critical.

However, its therapeutic potential is limited by low stability and poor bioavailability. For berberine to exert its effects, it must achieve and maintain effective concentrations in circulation when taken orally. Nanotechnological approaches, which improve stability and bioavailability, represent a promising solution. Despite their benefits, these methods face challenges such as high production costs, scalability issues, and regulatory hurdles. Advances in manufacturing techniques and cost-reduction strategies are essential for integrating nanotechnology-based therapies into routine clinical practice.

Future research should focus on developing new methods suitable for oral administration that enhance encapsulation efficiency and loading capacity. Additional clinical trials are needed to address the low bioavailability and insufficient toxicity data, which currently prevent the U.S. Food and Drug Administration (FDA) from classifying berberine as a drug. Comprehensive studies in diverse populations are crucial to fully establish berberine's efficacy, optimal dosage, and clinical application.

### Funding
The authors received no funding for this work.

### Competing Interests
The authors declare that they have no competing interests.

### Author Contributions
- Aslıhan Alpaslan Ağaçdiken conceived and designed the experiments, performed the experiments, analyzed the data, prepared figures and/or tables, authored or reviewed drafts of the article, and approved the final draft.
- Zeynep Göktaş conceived and designed the experiments, performed the experiments, analyzed the data, authored or reviewed drafts of the article, and approved the final draft.

### Data Availability
This is a literature review.

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
