# Peer review of "Berberine-induced browning and energy metabolism: mechanisms and implications"

_PeerJ, doi:10.7717/peerj.18924_

## Round 0.1 · original submission · Major Revisions

The article provides valuable insights into berberine's effects on browning. However, some points require clarification:
Structure:
• Missing Section: The article could benefit from a dedicated section on energy expenditure. Explain the concept of thermogenesis and mention its minimal contribution to overall body energy use.
• Two paragraphs, particularly those on browning mechanisms and BAT activation, and adenosine monophosphate-activated protein kinase (AMPK), are quite lengthy and information-dense. These sections could benefit from rewriting and synthesis to improve readability and clarity. The Browning Mechanisms and BAT Activation parapgprah can be summarized into a table for easier reference.
• The conclusion suddenly started to discuss commercial use when the entire review was about in-vivo. Human research was lacking in the review.
Clarity:
• Line 99-100: The distinction between oral and intragastric administration (do you mean by NG tube) is unclear. Consider clarifying if both methods were employed in the studies reviewed.
• Lines 106-112: This section could be condensed into one paragraph.
• Line 156: Brown adipose tissue (BAT) is not solely present in supraclavicular area. The statement gives this impression.
• Line 166: Specify the types of dietary changes mentioned as triggers for browning.
• Line 184: Rephrase for clarity. Perhaps: "β-adrenergic receptor activation is considered a key stimulus for browning..."
• Lines 238-242: This information seems repetitive; consider integrating it with the previous paragraphs addressing similar points (lines 177-182).
• Line 294: Add "is" before "negatively" for proper grammar.
• Line 374-370: Specify studies where Berbeine was used in dendrimers
• Line 383: Clarify the presence of in vitro (cell-based) studies alongside the in vivo (whole-organism) research mentioned earlier, especially that suddenly we have the word commercial form in the conclusion.
• Line 389: Maintain consistency. Use "a herbal product" instead of "an herbal" Overall:
This review provides valuable insights but could benefit from improved clarity and structure with the suggested edits.

Reviewer 1 ·

Basic reporting

Clear English used throughout, add “is” in line 295 after “differentiation factor 15 expression”
More recent articles can be cited e.g
-Gasmi A, Asghar F, Zafar S, Oliinyk P, Khavrona O, Lysiuk R, Peana M, Piscopo S, Antonyak H, Pen JJ, Lozynska I, Noor S, Lenchyk L, Muhammad A, Vladimirova I, Dub N, Antoniv O, Tsal O, Upyr T, Bjørklund G. Berberine: Pharmacological Features in Health, Disease and Aging. Curr Med Chem. 2024;31(10):1214-1234. doi: 10.2174/0929867330666230207112539. PMID: 36748808.
- Ilyas Z, Perna S, Al-Thawadi S, Alalwan TA, Riva A, Petrangolini G, Gasparri C, Infantino V, Peroni G, Rondanelli M. The effect of Berberine on weight loss in order to prevent obesity: A systematic review. Biomed Pharmacother. 2020 Jul;127:110137. doi: 10.1016/j.biopha.2020.110137. Epub 2020 Apr 27. PMID: 32353823.

Despite recent review articles on Berberine and weight loss, this review focuses mainly on the role Berberine plays in browning and discusses implications to improve bioavailability

The introduction could benefit from a discussion on Berberine safety issues, side effects, and contraindications if any. Also it would be important to discuss the role of browning, among all the different implicated Berberine effects, on weight loss.

Experimental design

It is clear that the authors did a rigorous review on the topic at hand
More details on number of articles retrieved from search in PubMed. Why were other search engines not utilized?
It would be beneficial to add typical doses of Berberine used in studies and expected weight loss in Kgs.

Validity of the findings

Discuss the cost and feasibility of nanotechnology and practical implications.

Reviewer 2 ·

Basic reporting

No comment.

Experimental design

no comment

Validity of the findings

no comment

Additional comments

1,This study first describes the metabolism of berberine, the basic classification and functional status of white and brown adipose tissue, followed by the mechanisms of browning and brown adipose tissue activation. Finally, it discusses the effects and pathways of berberine in brown adipose tissue activation, listing two key factors AMPK and GDF15, as well as a series of packaging methods and applicability of berberine. Finally, the conclusion is drawn. The review has a certain level of organization, but lacks innovation, Lack of eye-catching viewpoints and lacks the latest research progress. The research content is relatively scattered and the themes are not focused enough.

2, The abstract first mentioned the panic of obesity, increased energy consumption, reduced appetite, and the importance of brown adipose tissue. To stimulate the positive effects of berberine by consuming energy. However, in the Introduction (Line 34) of the main text, the source, composition, and efficacy of berberine are mentioned first. Then it was mentioned that there is a potential effect of weight loss, as it promotes glucose and lipid metabolism, inhibits and regulates microorganisms. Here, it does not obediently introduce the browning of adipose tissue, but stating that berberine can induce browning and thermogenesis of adipose tissue (Line 52-53), which seems a bit abrupt. The reason of linking brown fat to the discovery of thermogenic function of berberine is rather far-fetched.

3, The title is about the mechanism and implications of berberine induced browning of adipose tissue, but the overall of berberine metabolism and packaging methods is not specifically targeted at adipose tissue or brown adipose tissue, but applied to the whole body. Therefore, it is suggested that the author revise it slightly wider, such as energy metabolism of skeletal muscle, myocardium, fat, brain, etc., and develop some targeted packaging materials to delay metabolism, which can have a more meaningful effect after absorption. The author may consider refining the title or reorganizing the logical structure of the paper writing.

4, But in terms of mechanism, which is the key content. The author listed many methods and proteins that activate brown adipose tissue. Table 2 lists the doses and studied pathways of berberine intake in cells, rats, mice, and humans. The dose range is somewhat broad and the time period is long-term. Is there a summary of the lowest effective dose or other signaling pathways besides AMPK and GDF15? What dosage works through which pathways and what regulatory mechanisms are involved.

Actually, the mechanism is not very clear, and most of the literature is only observation of phenomena, not gene knockout followed by berberine supplementation to determine the target. As referenced in Wang et al. 2021; Yang et al. (2017) found that GDF15 binds to its receptor, increases UCP1 expression levels, promotes browning, and subsequently reduces body weight.

5, Some descriptions are not precise enough and are rather vague, L343-344, how much better is the bioavailability of berberine-SLN than free berberine? Please write it down, then it will look more convincing.

6, Is the nanotechnology methods of berberine specifically designed for brown adipose tissue? Or they are fit for the whole body.

7, There was also a lot of statement about the classification of adipose tissue, the sources and activation of brown fats, but in fact, many of them are unrelated to berberine directly.

8, When it comes to the regulation of brown adipose tissue by berberine through the AMPK and GDF15 pathways, there is a lot of talk about the regulation of AMPK, but it is actually not related to berberine. At last, I recommend the authors can carefully sort out the key points and logic of this study.

---

## Round 0.2 · Minor Revisions

Thank you for your revised manuscript and for addressing the reviewers' comments. I appreciate the effort you have put into improving the clarity and quality of your work.

However, there are a few remaining issues that need to be addressed:

Search Strategy and Publication Numbers: Please incorporate your response regarding the search strategy and the number of publications into the manuscript. This will enhance transparency and provide readers with a clear understanding of the scope of your literature review. Specifically, please include:

The number of articles you initially identified through your search.
The number of articles you ultimately included in your review.
The number of articles excluded and the reasons for exclusion (e.g., low impact factor, irrelevant topic, etc.).


Minor Textual Corrections:

Line 193: While you have modified the section on BAT in the supraclavicular region, further clarification is needed. Please revise the sentence to acknowledge that the first BAT in adults was found in the supraclavicular region. Follow this with a reference that reflects the current understanding of BAT distribution throughout the body.
Line 224: Please begin the sentence with "Twenty percent" instead of using the numeral.
Line 283: Delete the full stop after "Table 2."
Line 420: Ensure the accuracy and consistency of the reference citation. You can use an in-text citation or at the end of the sentence.

Please address these remaining points and resubmit your revised manuscript. I look forward to receiving your updated version soon.

---

## Round 0.3 · accepted · Accept

Thank you for your careful attention to the grammatical and editing suggestions. I have reviewed your revised manuscript and am happy with your changes.

I am pleased to inform you that it has now been accepted for publication in PeerJ.